# Identification of High-Order Nonlinear Coupled Systems Using a Data-Driven Approach

**Rodolfo Daniel Velázquez-Sánchez** [1,†] , **Jonathan Omega Escobedo-Alva** [2,†] , **Raymundo Peña-García** [1,†] , **Ricardo Tapia-Herrera** [3,†] **and Jesús Alberto Meda-Campaña** [1,*,†]

1   SEPI ESIME Zacatenco, Instituto Politécnico Nacional, Ciudad de México 07738, Mexico; rvelazquezs1901@alumno.ipn.mx (R.D.V.-S.); rpenag1600@alumno.ipn.mx (R.P.-G.)
2   SEPI ESIME Ticomán, Instituto Politécnico Nacional, Ciudad de México 07340, Mexico; jescobedoa@ipn.mx
3   SEPI ESIME Zacatenco, CONAHCYT—Instituto Politécnico Nacional, Ciudad de México 07738, Mexico; rtapiah@ipn.mx
*   Correspondence: jmedac@ipn.mx
†   These authors contributed equally to this work.

**Abstract:** Most works related to the identification of mathematical nonlinear systems suggest that such approaches can always be directly applied to any nonlinear system. This misconception is greatly discouraging when the obtained results are not expected. Thus, the current work hypothesizes that the more information one has about the mathematical structure of the model, the most precise the identification result. Therefore, a variant of the Sparse Identification of Nonlinear Dynamics (SINDY) approach is presented to obtain the full mathematical nonlinear model of a high-order system with coupled dynamics, namely, a commercial quadcopter. Furthermore, due to its high sensitivity to inputs, a control system is devised using the identified model to stabilize the quadcopter. This illustrates the effectiveness of the proposed identification method.

**Keywords:** Sparse Identification of Nonlinear Dynamics; high-order systems; coupled dynamics

## 1. Introduction

### 1.1. Motivation

Understanding the differential equations that govern the dynamics of nonlinear systems in real-time is always considered advantageous in scientific research. However, the availability of accurate mathematical models for such systems is not always guaranteed. Despite the recognized benefits of having access to these equations, practical limitations and inherent complexities often prevent the simple formulation of mathematical representations for real-world nonlinear systems. Consequently, many authors have focused their efforts on the practical identification of mathematical models for nonlinear systems [1].

On the other hand, the continuously growing capabilities of computing and data acquisition equipment have produced an explosion of data-driven models, which in many cases are more appealing than those obtained by analytical modeling approaches [2].

Ahead of the data-driven modeling techniques is Deep Learning (DL) [3,4]. DL has not only shown notable performance in image classification but is also very effective in predicting the future states of dynamical systems. However, its main drawback is the difficulty relating the DL models to the governing equations of the physical systems. A different data-driven approach is based on symbolic regression to obtain the structure of a nonlinear system from data [5]. Although this method identifies the interpretable physical models very well, its main drawback is the computational cost of working with a symbolic regressor, and it grows significantly as the dimension of the problem does. A genetic algorithm, considering the transition matrix in its fitness function, has been successfully applied to identify the physics-based longitudinal dynamics of an aircraft. However, its

main drawback is that it is an iterative process strongly dependent on the size of the search region for parameters [6].

In [7], the problem has been restated as sparse identification combined with the Levenberg–Marquardt (LM) algorithm. Tibshirani first introduced the main idea for robust identification, which combines least squares minimization and discrimination through the $L_1$ norm, and named it Shrinkage Lasso Selection [8]. However, the problem has been simplified even more by the Sparse Identification of Nonlinear Dynamics (SINDY), which combines sparse regression with the advantages of the verified efficacy of the least squares (LS) approach [9,10].

Brunton et al. [9] proposed a method called Sparse Identification of Non-Linear Dynamics in 2016 which transforms a nonlinear model of the form $\dot{x} = f(x, u)$, where $x \in \mathbb{R}^n$ is the state vector and $u \in \mathbb{R}^p$ is the input vector, into an equivalent linear model of the form $\dot{X} = \Theta(X, U)\Xi$ (please refer to Section 2.2 for a more detailed explanation of this concept). This simplifies the nonlinear identification problem to solving a linear system with more equations than unknowns, which is a classical problem in linear algebra, namely, $Ax = b$. This method offers an elegant approach to identifying nonlinear dynamics from data. The method can handle complex systems with perfect data from ideal experiments, as demonstrated in [9]. However, for simulations with imperfect data, the method requires some modifications, which are explained in [9].

The robustness analysis of the method with respect to the choice of "hyperparameters" is presented in [11]. The hyperparameters are related to the Lasso-type optimization, which is a classical optimization problem that uses the $L_1$ norm instead of the $L_2$ norm, as described in [12]. The extension of the method to systems with control inputs is discussed in [9]. SINDY has been applied during the identification of the models of complex systems as chaotic attractors, even in the presence of control signals, in a very successful way [9].

However, the original SINDY algorithm proves inadequate for high-order nonlinear systems with coupled dynamics, even when the structure of the mathematical model is known. The limitation arises from the reliance on the least squares approach, favoring the selection of large values over small. Consequently, the SINDY algorithm may exclude functions with small coefficients in nonlinear models where coefficients vary significantly in magnitude. This can result in inaccurate or incomplete models that fail to capture the true dynamics of the system. For instance, in a recent work [13], the authors proposed a modification of the SINDY algorithm, aiming to outperform the classical SINDY approach in constructing a system of ordinary differential equations from stochastic simulations.

Thus, despite the simplicity of SINDY, it requires constant modifications and improvements tailored to each new application. As a result, customized variants of the SINDY algorithm have begun to emerge in the recent literature [14–18].

### 1.2. Contribution

With this in mind, in the context of a known structure for a high-order nonlinear system with coupled dynamics, this work presents a significant contribution by introducing a simple modification of the SINDY methodology. Specifically, this modification allows the identification of missing coefficients within the known system structure. What distinguishes this contribution is not only its effectiveness but also its minimal computational cost. By seamlessly integrating into the existing SINDY framework, this modification offers a practical solution to a challenging problem, improving the applicability and accuracy of the method without imposing significant computational overhead. Therefore, in addition to improving SINDY's capabilities for system identification, this work also underlines the importance of efficiency in algorithmic improvements within the field of nonlinear dynamical systems [19,20]. A comparison between the performance of the original algorithm and modified one was carried out by applying both to the identification of the mathematical model of a commercial quadcopter.

*1.3. Manuscript Organization*

The rest of this paper is organized as follows: In Section 2, the mathematical and numerical tools considered throughout the study are briefly reviewed. In Section 3, the classical SINDY algorithm is employed to identify the high-order nonlinear system with coupled dynamics. In Section 4, the modified algorithm is introduced and employed for the identification of the same system. In Section 5, some important points are discussed. Finally, Section 6 presents the concluding remarks.

## 2. Mathematical Tools

*2.1. LS*

The least squares method was introduced by Adrien-Marie Legendre in 1805 [21]. Roughly speaking, the least squares solution for the problem $Ax = b$, where $A \in \mathbf{R}^{m \times n}$, $x \in \mathbf{R}^n$, and $b \in \mathbf{R}^m$, is a vector $\hat{x}$ such that $\mathrm{dist}(b, A\hat{x}) \leq \mathrm{dist}(b, Ax)$, for all other vectors $x \in \mathbf{R}^n$, where $\mathrm{dist}(b, A\hat{x}) = \|b - A\hat{x}\|$ is the square root of the sum of the squares of the elements of the vector $b - A\hat{x}$. Therefore, a least squares solution minimizes the sum of the squares of the errors between $b$ and $A\hat{x}$, hence its name [22].

When the columns of $A$ are linearly independent, the least squares solution of

$$Ax = b \tag{1}$$

is unique and it is given by [22]:

$$\hat{x} = (A^T A)^{-1} A^T b. \tag{2}$$

Equivalently, if the $n$ columns of $A$, namely, $a_1, a_2, \ldots, a_n$, are orthogonal, then the least squares solution of (1) is [22]:

$$\hat{x} = \left( \frac{a_1 \cdot b}{a_1 \cdot a_1}, \frac{a_2 \cdot b}{a_2 \cdot a_2}, \ldots, \frac{a_n \cdot b}{a_n \cdot a_n} \right). \tag{3}$$

Moreover, the *QR* factorization can be applied to the matrix $A$, and the Gram–Schmidt algorithm can be used to obtain the orthogonal $Q$ and upper triangular $R$ matrices, which can be used to obtain the least squares solution of (1). For a thorough analysis of the least squares method, the *QR* factorization and the Gram–Schmidt algorithm, the reader can refer to [22,23].

*2.2. SINDY*

In 2016, Brunton et al. [9] introduced Sparse Identification of Nonlinear Dynamics (SINDY), which consists of the simple idea of solving the classical linear problem $Ax = b$ but transforming it into

$$\dot{X} = \Theta(X, U)\Xi, \tag{4}$$

where the nonlinear model to be estimated has the form $\dot{x} = f(x, u)$, with $x \in \mathbf{R}^n$ and $u \in \mathbf{R}^p$. In this formulation, $X \in \mathbf{R}^{m \times n}$, where the i-th column of $X$ corresponds to a vector formed by the $m$ samples of the i-th state of $x$, the same applies for $\dot{X}$, and $U \in \mathbf{R}^{m \times p}$, where the i-th column of $U$ corresponds to a vector formed by the $m$ samples of the i-th control input of $u$. On the other hand, $\Theta(X, U) \in \mathbf{R}^{m \times q}$, where each column of $\Theta(X, U)$ is a candidate function depending on $x$ and/or $u$, which is evaluated at each one of the m sample instants, and the columns of $\Xi \in \mathbf{R}^{q \times n}$ are sparse vectors of coefficients.

The goal is to find the matrix $\Xi$ by means of least squares such that the residual error is minimized. The $\Xi$ is a matrix of coefficients that determine the active terms in the nonlinear model $\dot{x} = f(x, u)$, which can be approximated by a linear combination of candidate functions depending on $x$ and/or $u$. The pseudocode for SINDY and its interpretation are presented in Appendix A.1 [9,12].

### 2.3. Mathematical Model of a Quadcopter

A quadcopter or quadrotor is a type of unmanned aerial vehicle (UAV) that has four rotors mounted on a rigid frame. The rotors can generate both lift and torque, which are used to control the position and orientation of the quadrotor. The mathematical model of a quadrotor can be derived using Newtonian and Euler's laws and applying basic principles of physics. This derivation gives the equations that govern the motion of a quadrotor, both concerning the body frame and the inertial frame.

The configuration space of a quadcopter is defined by six variables: $x$, $y$, $z$, $\phi$, $\theta$, and $\psi$, where $(x, y, z)$ are the coordinates of the center of mass in an inertial frame, and $(\phi, \theta, \psi)$ are the Euler angles that represent the orientation of the body frame with respect to the inertial frame. The actuation space of a quadrotor is defined by four variables: $u_1$, $u_2$, $u_3$, and $u_4$, where $u_i$ is the angular velocity of the i-th rotor. The schematics of this aircraft are given in Figure 1. In aeronautics, the convention of measuring elevation or altitude through the negative part of the Z-axis is often a matter of mathematical and geometric consistency. The convention is motivated by the right-hand coordinate system commonly used in aviation and engineering: the X-axis points forward (along the longitudinal axis of the aircraft), the Y-axis points to the right (along the lateral axis), and the Z-axis points downward (along the vertical axis) [24].

The dynamics of a quadcopter can be expressed as [25–28]:

$$\dot{x} = f(x, u), \tag{5}$$

where $x = [x_1 \ldots x_{12}]^T$, $u = [u_1 \ \ldots \ u_4]^T$, and

$$\dot{x}_1 = x_7, \tag{6}$$

$$\dot{x}_2 = x_8, \tag{7}$$

$$\dot{x}_3 = x_9, \tag{8}$$

$$\dot{x}_4 = x_{12}, \tag{9}$$

$$\dot{x}_5 = x_{11}, \tag{10}$$

$$\dot{x}_6 = x_{10}, \tag{11}$$

$$\dot{x}_7 = \frac{\beta_1}{m}(\sin(x_4)\sin(x_6) + \cos(x_4)\cos(x_6)\sin(x_5)), \tag{12}$$

$$\dot{x}_8 = -\frac{\beta_1}{m}(\cos(x_4)\sin(x_6) - \cos(x_6)\sin(x_4)\sin(x_5)), \tag{13}$$

$$\dot{x}_9 = \frac{\beta_1}{m}(\cos(x_5)\cos(x_6)) + 9.81, \tag{14}$$

$$\dot{x}_{10} = \frac{1}{I_{xx}}(\beta_2 + (I_{yy} - I_{zz})x_{11}x_{12}), \tag{15}$$

$$\dot{x}_{11} = \frac{1}{I_{yy}}(\beta_3 + (I_{zz} - I_{xx})x_{10}x_{12}), \tag{16}$$

$$\dot{x}_{12} = \frac{1}{I_{zz}}(\beta_4 + (I_{xx} - I_{yy})x_{10}x_{11}), \tag{17}$$

with $\beta_1$ as the force responsible for throttle movement, $\beta_2$ as the torque responsible for roll movement, $\beta_3$ as the torque responsible for pitch movement, and $\beta_4$ as the torque responsible for yaw movement, given by

$$\beta_1 = b(u_1^2 + u_2^2 + u_3^2 + u_4^2), \tag{18}$$

$$\beta_2 = b(u_4^2 + u_3^2 - u_1^2 - u_2^2), \tag{19}$$

$$\beta_3 = b(u_2^2 + u_3^2 - u_1^2 - u_4^2), \tag{20}$$

$$\beta_4 = d(u_1^2 + u_3^2 - u_2^2 - u_4^2), \text{ and} \tag{21}$$

$$\Omega = u_1 - u_2 + u_3 - u_4. \tag{22}$$

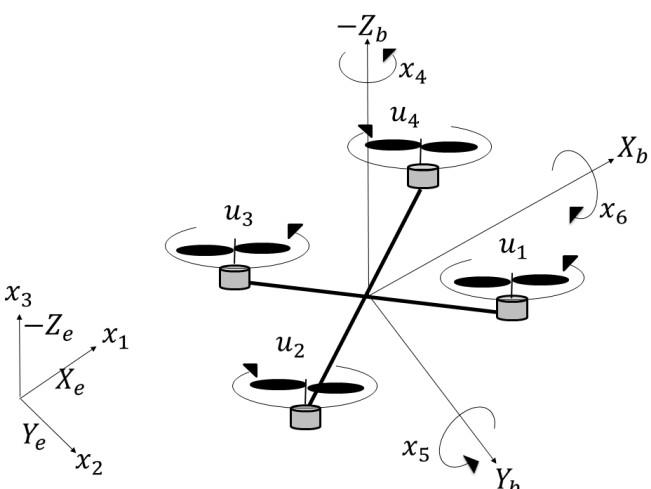

**Figure 1.** Body diagram of the quadcopter.

Notice that $u_1$, $u_2$, $u_3$, and $u_4$ are the frequencies of rotors 1, 2, 3 and 4, respectively, i.e., they are the implicit control inputs and they are given in rad/s . The variables $x_1$, $x_2$, and $x_3$ represent the linear displacements along the earth fixed axes $X_e$, $Y_e$, and $Z_e$, respectively, and they are in meters. On the other hand, the variables $x_4$, $x_5$, and $x_6$ describe the angular displacements around the body axes $Z_b$, $Y_b$, and $X_b$, respectively, and they are in radians. The remaining state variables describe the linear and angular velocities, and they can be easily deduced from (5)–(22).

### 2.4. Simulink Support Package for Parrot Minidrones

The Simulink Support Package for Parrot Minidrones version 20.2.3 is a toolbox that enables users to design and deploy flight control algorithms for Parrot minidrones using Simulink. Users can wirelessly connect to the minidrones via Bluetooth and access their onboard sensors and camera. Users can also use additional tools such as the Aerospace Blockset and Simulink Coder to enhance their simulations and code generation. The toolbox also includes an example that shows how to model and simulate the 6-DOF equations of motion for the minidrones [29,30]. In Figure 2, the physical quadcopter simulated by the Simulink Support Package for Parrot Minidrones is depicted.

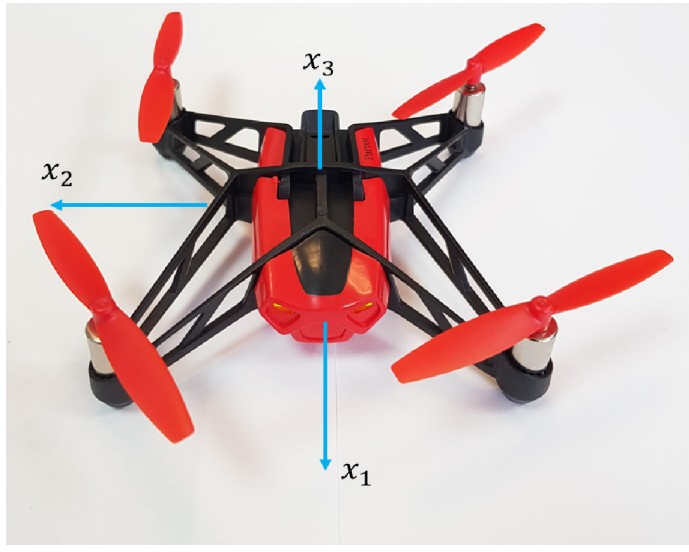

**Figure 2.** Rolling Spider by Parrot Minidrones.

This package includes a Controller Project template designed as a framework for crafting a personalized controller or adapting it to specific needs. The template integrates a robust plant model simulation for both the Parrot Rolling Spider and Parrot Mambo drones. This simulation streamlines the evaluation of model outcomes before deployment, empowering users to scrutinize and refine their controller designs prior to applying them to the physical hardware. The template supports the modeling of equations of motion with six degrees of freedom (6-DOF), facilitating the simulation of aircraft behavior under diverse flight and environmental conditions. Figure 3 depicts the VRML environment employed by the Simulink Package for Parrot Minidrones.

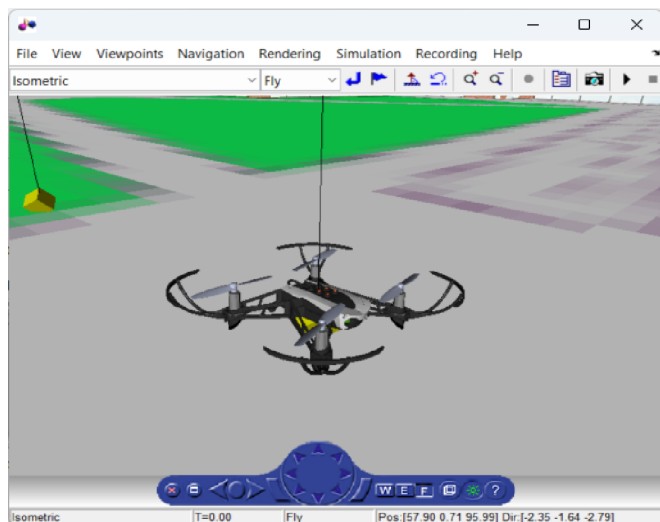

**Figure 3.** Capture of the VRML environment provided by the Simulink Support Package for Parrot Minidrones.

It is worth mentioning that the Simulink Support Package for Parrot Minidrones has undergone extensive validation, confirming its high accuracy in approximating the real-time behavior of the Parrot Minidrones. As a result, the identification of the quadcopter's mathematical model in this study relies heavily on the results generated by the Simulink package.

### 3. Identification of a High-Order Nonlinear Systems with Coupled Dynamics Using the Classical SINDY Algorithm

In this section, the Algorithm A1 is used to obtain the mathematical model of a quadcopter from the data generated by the Simulink Support Package for Parrot Minidrones.

For the sake of simplicity, throughout the remainder of this study, it is assumed that the Rolling Spider MiniDrone depicted in Figure 2 can be effectively modeled using Equations (5)–(17). In other words, the drone's controls are considered to be $\beta_1$, $\beta_2$, $\beta_3$, and $\beta_4$, instead of $u_1$, $u_2$, $u_3$, and $u_4$. Adopting this perspective streamlines the required library of functions for SINDY, specifically the matrix $\Theta(X, U)$. Despite this simplification, the resulting nonlinear system maintains sufficient complexity to reveal the limitations of the conventional SINDY approach. This underscores the advantages of the proposed modification, particularly in the identification of high-order nonlinear systems with coupled dynamics.

The Simulink Support Package for Parrot Minidrones is employed to simulate a 10-s flight sequence of the drone, with the sampling time $T = 0.005\,\text{s}$. Throughout this duration, the drone undergoes takeoff and ascends to an altitude of 1.5 m. Meanwhile, the controls $\beta_1$, $\beta_2$, $\beta_3$, and $\beta_4$ are subtly perturbed to excite all four independent degrees of freedom of the drone. These inputs are shown in Figure 4.

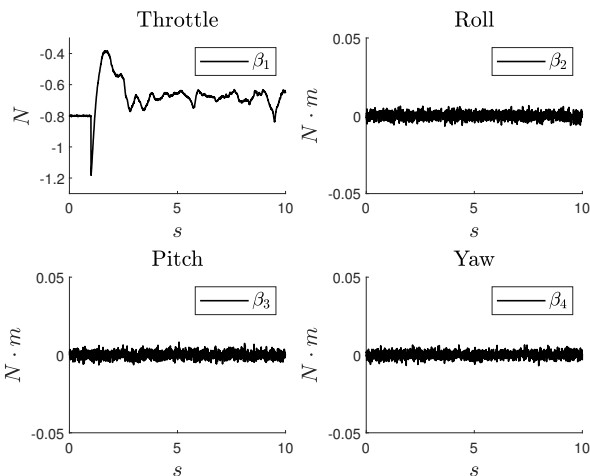

**Figure 4.** Input signals used to excite the four degrees of freedom of the drone.

During this simulated flight, the vectors $x_1, \ldots, x_{12}, \dot{x}_1, \ldots, \dot{x}_{12}, \beta_1, \ldots, \beta_4 \in \mathbf{R}^m$ are formed by reading the linear and angular displacements, linear and angular velocities, linear and angular accelerations, and controls. Clearly, $m = 2001$.

So, to correctly formulate Equation (4), consider that the matrix $\dot{X}$ is:

$$\dot{X} = [\dot{x}_1, \ldots, \dot{x}_{12}], \tag{23}$$

while the matrix $\Theta(X, U)$ is

$$\Theta(X, U) = [f_1\, f_2\, \cdots\, f_{16}], \tag{24}$$

with $f_1 = 1$, $f_2 = x_7$, $f_3 = x_8$, $f_4 = x_9$, $f_5 = x_{10}$, $f_6 = x_{11}$, $f_7 = x_{12}$, $f_8 = \beta_1 \cdot (\sin(x_4) \cdot \sin(x_6) + \cos(x_4) \cdot \cos(x_6) \cdot \sin(x_5))$, $f_9 = \beta_1 \cdot (\cos(x_4) \cdot \sin(x_6) - \cos(x_6) \cdot \sin(x_4) \cdot \sin(x_5))$, $f_{10} = \beta_1 \cdot \cos(x_5) \cdot \cos(x_6)$, $f_{11} = \beta_2$, $f_{12} = x_{11} \cdot x_{12}$, $f_{13} = \beta_3$, $f_{14} = x_{10} \cdot x_{12}$, $f_{15} = \beta_4$, $f_{16} = x_{10} \cdot x_{11}$, where, as mentioned above, $\dot{x}_1, \ldots, \dot{x}_{12}, f_1, \ldots, f_{16} \in \mathbf{R}^{2001}$. Thus, the unknown matrix $\Xi$ has a dimension of $16 \times 12$.

The MATLAB code for the classical SINDY algorithm, with *Theta* $= \Theta(X, U)$, $dXdt = \dot{X}$, *lambda* $= 1 \times 10^{-3}$, and $n = 12$, is given in Appendix A.2.

The matrix $\Xi$ produced is:

$$\Xi = \begin{bmatrix} 0 & 0 & 0 & 0 & 0 & 0 \\ 1 & 0 & 0 & 0 & 0 & 0 \\ 0 & 1 & 0 & 0 & 0 & 0 \\ 0 & 0 & 1 & 0 & 0 & 0 \\ 0 & 0 & 0 & 0 & 0 & 1 \\ 0 & 0 & 0 & 0 & 1 & 0 \\ 0 & 0 & 0 & 1 & 0 & 0 \\ 0 & 0 & 0 & 0 & 0 & 0 \\ 0 & 0 & 0 & 0 & 0 & 0 \\ 0 & 0 & 0 & 0 & 0 & 0 \\ 0 & 0 & 0 & 0 & 0 & 0 \\ 0 & 0 & 0 & 0 & 0 & 0 \\ 0 & 0 & 0 & 0 & 0 & 0 \\ 0 & 0 & 0 & 0 & 0 & 0 \\ 0 & 0 & 0 & 0 & 0 & 0 \\ 0 & 0 & 0 & 0 & 0 & 0 \end{bmatrix}$$

$$\begin{bmatrix}
-0.01 & 0.18 & 4.8 & 1.1 & 1.1 & -0.076 \\
0.21 & 0.2 & 0.13 & 1.2 & 5.6 & -0.053 \\
9.1 \times 10^{-3} & 0.018 & -1.5 \times 10^{-3} & -0.12 & -2.4 & 0.095 \\
0.051 & 2.3 \times 10^{-3} & -0.096 & 0.63 & -1.3 & -0.21 \\
0.034 & -0.14 & 0.093 & -0.67 & -1.2 & -0.4 \\
0.12 & -0.071 & 0.11 & -0.12 & -3.8 & 0.12 \\
0.072 & 0.064 & 0.18 & 1.7 & -0.14 & -1.1 \\
15.0 & 1.3 & -0.1 & -8.3 & -8.5 & -2.0 \\
-0.066 & -14.0 & 1.0 & -5.5 & 1.6 & -1.1 \\
-0.015 & 0.27 & 7.5 & 2.0 & 2.9 & 0.11 \\
1.1 & 0.52 & 5.7 & 8.2 \times 10^3 & -90.0 & -6.7 \\
-0.068 & -0.027 & -0.054 & -2.5 & -6.2 & -0.46 \\
-0.77 & -1.1 & 6.4 & 2.0 & 5.9 \times 10^3 & 84.0 \\
0.033 & 0.13 & 0.041 & 4.5 & -9.6 & 0.48 \\
2.4 & 2.3 & 6.4 & 30.0 & 39.0 & 4.1 \times 10^3 \\
-0.048 & 0.069 & -0.073 & 1.7 & -2.2 & 0.2
\end{bmatrix} \tag{25}$$

It is noteworthy that the obtained identification does not exhibit sparsity, as each function considered in $\Theta(X, U)$ emerges between the seventh and twelfth equations in the identified model, even when the parameter lambda is not very small (e.g., $1 \times 10^{-3}$). However, even if one were to consider the identified model as a potential representation of the nonlinear dynamics of the drone, a significant limitation becomes apparent: the resulting model is non-controllable. This poses a fundamental issue, given that drones are recognized as controllable systems in the literature [31]. This discrepancy underscores the need for further refinement in the modeling approach to ensure an accurate representation of the controllable nature inherent to drone dynamics.

This example demonstrates that, in general, it is not sufficient to know the functions appearing in the model to be identified for SINDY to yield correct results. The following section addresses and mitigates this drawback.

## 4. Identification of a High-Order Nonlinear Systems with Coupled Dynamics through the Modified SINDY Algorithm

### 4.1. GenesisXi Algorithm

At this point, it becomes clear that simply knowing the functions within the model is insufficient. It is also crucial to determine which differential equation each function should correspond to. This shift from model identification to coefficient (parameter) identification is significant, particularly for real-world nonlinear problems.

As a result, modifying the traditional SINDY algorithm involves integrating this additional information. The goal is to refine the results obtained so that they closely resemble known models, such as those from quadrotors, thus improving the applicability and accuracy of the algorithm in real-world scenarios.

To achieve this objective, it is necessary to formulate an initial matrix $\Xi_0$ wherein the positions of all candidate functions contained in $\Theta(X, U)$ within the differential equations of the mathematical model are defined. To automate this process, namely, obtaining the aforementioned matrix $\Xi_0$, the GenesisXi algorithm is proposed. The pseudocode for this algorithm is described in Appendix A.3, while the corresponding MATLAB function, with *Theta_L* $= \Theta_L$, is given in Appendix A.4.

Thus, after applying GenesisXi algorithm, and in accordance with (5)–(17), the resulting matrix $\Xi_0$ is:

$$
\Xi_0 = \begin{bmatrix}
0 & 0 & 0 & 0 & 0 & 0 & 0 & 0 & 1 & 0 & 0 & 0 \\
1 & 0 & 0 & 0 & 0 & 0 & 0 & 0 & 0 & 0 & 0 & 0 \\
0 & 1 & 0 & 0 & 0 & 0 & 0 & 0 & 0 & 0 & 0 & 0 \\
0 & 0 & 1 & 0 & 0 & 0 & 0 & 0 & 0 & 0 & 0 & 0 \\
0 & 0 & 0 & 0 & 0 & 1 & 0 & 0 & 0 & 0 & 0 & 0 \\
0 & 0 & 0 & 0 & 1 & 0 & 0 & 0 & 0 & 0 & 0 & 0 \\
0 & 0 & 0 & 1 & 0 & 0 & 0 & 0 & 0 & 0 & 0 & 0 \\
0 & 0 & 0 & 0 & 0 & 0 & 1 & 0 & 0 & 0 & 0 & 0 \\
0 & 0 & 0 & 0 & 0 & 0 & 0 & 1 & 0 & 0 & 0 & 0 \\
0 & 0 & 0 & 0 & 0 & 0 & 0 & 0 & 1 & 0 & 0 & 0 \\
0 & 0 & 0 & 0 & 0 & 0 & 0 & 0 & 0 & 1 & 0 & 0 \\
0 & 0 & 0 & 0 & 0 & 0 & 0 & 0 & 0 & 1 & 0 & 0 \\
0 & 0 & 0 & 0 & 0 & 0 & 0 & 0 & 0 & 0 & 1 & 0 \\
0 & 0 & 0 & 0 & 0 & 0 & 0 & 0 & 0 & 0 & 1 & 0 \\
0 & 0 & 0 & 0 & 0 & 0 & 0 & 0 & 0 & 0 & 0 & 1 \\
0 & 0 & 0 & 0 & 0 & 0 & 0 & 0 & 0 & 0 & 0 & 1
\end{bmatrix}. \tag{26}
$$

Notice that the columns of $\Xi_0$ represent the differential equations in the nonlinear model, while the rows of $\Xi_0$ represent the candidate functions included in $\Theta(X, U)$. Thus, the matrix (26) indicates that $f_1$ appears in the ninth differential equation, $f_2$ appears only in the second differential equation, and so forth.

### 4.2. Modified SINDY Algorithm

Now, the location of the candidate functions in the matrix $\Xi_0$ is passed as a parameter to the modified SINDY function, as illustrated in the MATLAB code presented in Appendix A.5.

The matrix $\Xi$ obtained by applying the MATLAB function provided in Appendix A.5 to the data described in the previous section is:

$$
\Xi = \begin{bmatrix}
0 & 0 & 0 & 0 & 0 & 0 \\
1 & 0 & 0 & 0 & 0 & 0 \\
0 & 1 & 0 & 0 & 0 & 0 \\
0 & 0 & 1 & 0 & 0 & 0 \\
0 & 0 & 0 & 0 & 0 & 1 \\
0 & 0 & 0 & 0 & 1 & 0 \\
0 & 0 & 0 & 1 & 0 & 0 \\
0 & 0 & 0 & 0 & 0 & 0 \\
0 & 0 & 0 & 0 & 0 & 0 \\
0 & 0 & 0 & 0 & 0 & 0 \\
0 & 0 & 0 & 0 & 0 & 0 \\
0 & 0 & 0 & 0 & 0 & 0 \\
0 & 0 & 0 & 0 & 0 & 0 \\
0 & 0 & 0 & 0 & 0 & 0 \\
0 & 0 & 0 & 0 & 0 & 0 \\
0 & 0 & 0 & 0 & 0 & 0
\end{bmatrix}
$$

$$
\begin{bmatrix}
0 & 0 & 4.725 & 0 & 0 & 0 \\
0 & 0 & 0 & 0 & 0 & 0 \\
0 & 0 & 0 & 0 & 0 & 0 \\
0 & 0 & 0 & 0 & 0 & 0 \\
0 & 0 & 0 & 0 & 0 & 0 \\
0 & 0 & 0 & 0 & 0 & 0 \\
0 & 0 & 0 & 0 & 0 & 0 \\
14.32 & 0 & 0 & 0 & 0 & 0 \\
0 & -14.29 & 0 & 0 & 0 & 0 \\
0 & 0 & 7.302 & 0 & 0 & 0 \\
0 & 0 & 0 & 8169.0 & 0 & 0 \\
0 & 0 & 0 & -2.442 & 0 & 0 \\
0 & 0 & 0 & 0 & 5939 & 0 \\
0 & 0 & 0 & 0 & -7.961 & 0 \\
0 & 0 & 0 & 0 & 0 & 4087 \\
0 & 0 & 0 & 0 & 0 & 0.09223
\end{bmatrix}. \tag{27}
$$

Unlike the matrix presented in (25), this matrix corresponds to the sparse identification of a high-order nonlinear system characterized by coupled dynamics, specifically exemplified in the context of drone dynamics. Then, the identified nonlinear system is given by:

$$
\begin{align}
\dot{x}_1 &= x_7 \tag{28} \\
\dot{x}_2 &= x_8 \tag{29} \\
\dot{x}_3 &= x_9 \tag{30} \\
\dot{x}_4 &= x_{12} \tag{31} \\
\dot{x}_5 &= x_{11} \tag{32} \\
\dot{x}_6 &= x_{10} \tag{33} \\
\dot{x}_7 &= 14.32\beta_1(\sin(x_4)\sin(x_6) + \cos(x_4)\cos(x_6)sin(x_5)) \tag{34} \\
\dot{x}_8 &= -14.29\beta_1(\cos(x_4)\sin(x_6) - \cos(x_6)\sin(x_4)\sin(x_5)) \tag{35} \\
\dot{x}_9 &= 7.302\beta_1\cos(x_5)\cos(x_6) + 4.725 \tag{36} \\
\dot{x}_{10} &= 8169\beta_2 - 2.442x_{11}x_{12} \tag{37} \\
\dot{x}_{11} &= 5939\beta_3 - 7.961x_{10}x_{12} \tag{38} \\
\dot{x}_{12} &= 4087.0\beta_4 + 0.09223x_{10}x_{11}. \tag{39}
\end{align}
$$

Notice that the gravity value in Equation (36) is calculated as 4.725, which is 2.076 times smaller than 9.81. This implies that the entire equation is scaled by $2.076^{-1}$. However, as per Equations (12)–(14), when the identified model is analyzed at the equilibrium, i.e., when all linear and angular displacements, velocities, and accelerations are zero, and the mass $m$ is computed using Equations (34)–(36) multiplied by 2.076, it results in $m = 0.0698$, $m = 0.0699$, and $m = 0.0659$, respectively, which are very close to the nominal value found in the literature of $m = 0.068$ [29,30].

Now, based on these equations, a linear stabilizer is designed. To achieve this, Equations (28)–(39) are linearized at the equilibrium $x_{eq} = [0\ 0\ 0\ 0\ 0\ 0\ 0\ 0\ 0\ 0\ 0\ 0]^T$, resulting in:

$$
\dot{x} = Ax + Bu, \tag{40}
$$

with

$$
A = \begin{bmatrix}
0 & 0 & 0 & 0 & 0 & 0 & 1 & 0 & 0 & 0 & 0 & 0 \\
0 & 0 & 0 & 0 & 0 & 0 & 0 & 1 & 0 & 0 & 0 & 0 \\
0 & 0 & 0 & 0 & 0 & 0 & 0 & 0 & 1 & 0 & 0 & 0 \\
0 & 0 & 0 & 0 & 0 & 0 & 0 & 0 & 0 & 0 & 0 & 1 \\
0 & 0 & 0 & 0 & 0 & 0 & 0 & 0 & 0 & 0 & 1 & 0 \\
0 & 0 & 0 & 0 & 0 & 0 & 0 & 0 & 0 & 1.0 & 0 & 0 \\
0 & 0 & 0 & 0 & -9.27 & 0 & 0 & 0 & 0 & 0 & 0 & 0 \\
0 & 0 & 0 & 0 & 0 & 9.247 & 0 & 0 & 0 & 0 & 0 & 0 \\
0 & 0 & 0 & 0 & 0 & 0 & 0 & 0 & 0 & 0 & 0 & 0 \\
0 & 0 & 0 & 0 & 0 & 0 & 0 & 0 & 0 & 0 & 0 & 0 \\
0 & 0 & 0 & 0 & 0 & 0 & 0 & 0 & 0 & 0 & 0 & 0 \\
0 & 0 & 0 & 0 & 0 & 0 & 0 & 0 & 0 & 0 & 0 & 0
\end{bmatrix}, \tag{41}
$$

and

$$
B = \begin{bmatrix}
0 & 0 & 0 & 0 & 0 & 0 & 0 & 0 & 7.302 & 0 & 0 & 0 \\
0 & 0 & 0 & 0 & 0 & 0 & 0 & 0 & 0 & 8169 & 0 & 0 \\
0 & 0 & 0 & 0 & 0 & 0 & 0 & 0 & 0 & 0 & 5939 & 0 \\
0 & 0 & 0 & 0 & 0 & 0 & 0 & 0 & 0 & 0 & 0 & 4087
\end{bmatrix}^{T}. \tag{42}
$$

The stabilizer $u = -Kx$ is obtained using the Linear Quadratic Regulator (LQR) approach, which combines the control energy and the state deviations in the cost function of the form:

$$
J = \int_0^{\infty} \left( x^T Q x + u^T R u \right) dt, \tag{43}
$$

where $Q$ and $R$ are positive definite matrices determining the compromise between minimizing the state deviations and the control energy [31]. In this case, $Q = I_{[12 \times 12]}$, and $R = 1 \times 10^6 \cdot I_{[4 \times 4]}$. Thus, with matrices $A$, $B$, $Q$, and $R$ as above, the gain $K$ is computed using the MATLAB function *lqr*, resulting in:

$$
K = \begin{bmatrix}
0 & 0 & 0.001 & 0 & 0 & 0 \\
0 & 0.001 & 0 & 0 & 0 & 0.006795 \\
-0.001 & 0 & 0 & 0 & 0.007291 & 0 \\
0 & 0 & 0 & 0.001 & 0 & 0
\end{bmatrix}
$$
$$
\begin{bmatrix}
0 & 0 & 0.01658 & 0 & 0 & 0 \\
0 & 0.001571 & 0 & 0.001632 & 0 & 0 \\
-0.001604 & 0 & 0 & 0 & 0.001859 & 0 \\
0 & 0 & 0 & 0 & 0 & 0.00122
\end{bmatrix} \tag{44}
$$

Then, the stabilizer $u = -Kx$ is applied to the nonlinear model within the Simulink Support Package for Parrot Minidrones to assess the effectiveness of the identified model. In this example, the quadcopter reaches an altitude of 1 m using the linear stabilizer. The numerical results are presented in Figures 5 and 6. Figure 5 illustrates the drone's response to the linear stabilizer, while Figure 6 displays the corresponding input signals.

Please note that the goal of this work is not to design sophisticated controllers for complex drone behavior. Instead, the primary contribution of this work is to introduce an alternative method for obtaining the mathematical model of a high-order nonlinear system with coupled dynamics based on real-time measured data.

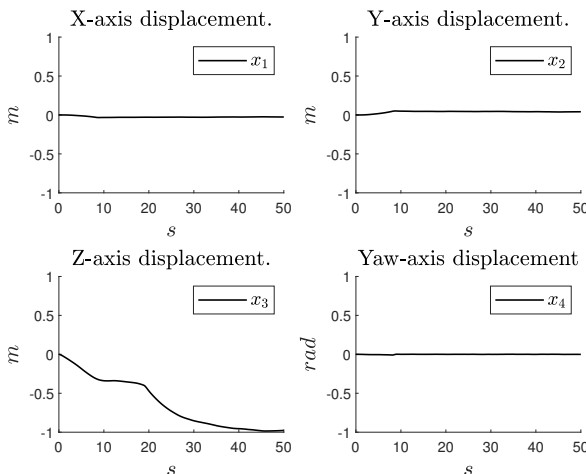

**Figure 5.** Quadcopter's four degrees of freedom with the linear stabilizer.

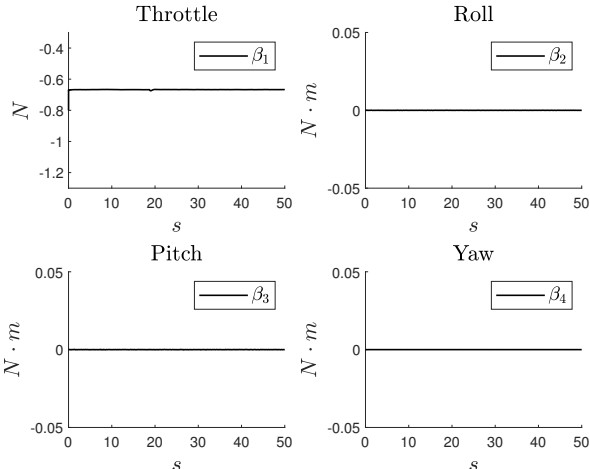

**Figure 6.** Input signals produced by the linear stabilizer.

### 5. Discussion

1.  In the specific case study, the outcomes achieved through the classical SINDY algorithm do not align with physics-based identification. Moreover, they do not qualify as sparse identification, as all candidate functions are present in the matrix $\Xi$.
2.  Based on the knowledge of the structure of the complex system to be identified, a more precise algorithm has been introduced without a significant increase in computational cost.
3.  Although the identified model appears to have significant differences from the related model found in the literature, it enables the design of a controller capable of stabilizing the original high-order nonlinear system with coupled dynamics. This, in turn, demonstrates its notable approximation degree.
4.  The primary drawback of the proposed approach is that the identification problem transforms into a coefficient identification task. Nevertheless, in many real-world applications, this outcome may still be appealing because, in numerous cases, it is impossible to precisely determine the coefficients of a specific system, even when the structure of its mathematical model is available.

### 6. Conclusions

This study presents a modification of the classical SINDY algorithm, facilitating the identification of mathematical models for high-order nonlinear systems with coupled dynamics. Despite its negligible computational cost compared to the benefits it offers, a key requirement is the prior knowledge of the nonlinear model's structure to generate the

template $\Xi_0$ for correctly placing the nonlinear functions within the respective differential equations. The efficacy of the proposed approach was confirmed through the application of a linear controller, designed based on the identified model, to the real system. Looking ahead, applying this approach to real-time quadcopter data, as opposed to the data provided by MATLAB, would be advantageous. However, this endeavor exceeds the current scope of the study.

**Author Contributions:** Conceptualization, R.D.V.-S. and J.A.M.-C.; formal analysis, R.P.-G., R.D.V.-S., J.O.E.-A. and R.T.-H.; investigation, J.O.E.-A., R.T.-H. and J.A.M.-C.; methodology, J.O.E.-A., R.T.-H. and J.A.M.-C.; project administration, J.A.M.-C.; writing—original draft, J.A.M.-C.; writing—review and editing, R.D.V.-S., J.O.E.-A., R.P.-G., R.T.-H. and J.A.M.-C. All authors have read and agreed to the published version of the manuscript.

**Funding:** This research received no external funding.

**Informed Consent Statement:** Not applicable.

**Data Availability Statement:** Data are contained within the article.

**Acknowledgments:** The authors gratefully acknowledge the partial support provided for this research by the Consejo Nacional de Humanidades Ciencias y Tecnologías (CONAHCYT) through the SNI (Sistema Nacional de Investigadores) scholarship. Additionally, the Instituto Politécnico Nacional contributed to this work through research projects 20230023 and 20240025, as well as by providing scholarships under the programs EDI (Estímulo al Desempeño de los Investigadores), COFAA (Comisión de Operación y Fomento de Actividades Académicas), and BEIFI (Beca de Estímulo Institucional de Formación de Investigadores).

**Conflicts of Interest:** The authors declare no conflicts of interest.

## Definitions

| | |
|---|---|
| $A$ | Coefficient matrix for the least squares problem. |
| $b$ | Vector of observed or measured values for the least squares problem. |
| $x$ | Vector of unknowns for the least squares problem. |
| $\hat{x}$ | Estimate or optimal solution for the least squares problem. |
| $X$ | Matrix whose columns are the states of the system, and the rows are the values of these states at different sampling instants. |
| $\dot{X}$ | Matrix whose columns are the first-time derivatives of the states of the system, and the rows are the values of such derivatives at different sampling instants. |
| $U$ | Matrix whose columns are the input signals of the system, and the rows are the values of such inputs at different sampling instants. |
| $\Theta(X, U)$ | Matrix whose columns constitute the library of candidate functions, dependent on the states and/or inputs of the nonlinear system. The rows represent the values of these functions at various sampling instants. |
| $\Xi$ | Sparse coefficient matrix that characterizes the importance or contribution of each term in the library of candidate functions (represented by $\Theta(X, U)$ ) to the dynamics of the system. |
| $\Xi_0$ | Template for $\Xi$, representing the structure of the nonlinear systems. |

## Appendix A. Pseudocodes and MATLAB Functions

*Appendix A.1. Classical SINDY Algorithm*

The following is a verbal description of Algorithm A1:

1. Given a series of snapshots $x \in \mathbf{R}^n$, $u \in \mathbf{R}^p$ and the corresponding time derivatives $\dot{x}$ of a dynamical system $\dot{x} = f(x, u)$, arrange them into matrices $X, \dot{X} \in \mathbf{R}^{m \times n}$, and $U \in \mathbf{R}^{m \times p}$ where $m$ is the number of samples, $n$ is the dimension of the state vector, and $p$ is the dimension of the input vector.

---

**Algorithm A1** SINDY algorithm

---

**Require:** Data matrix $X$, $U$, and derivative matrix $\dot{X}$
**Ensure:** Sparse model $\dot{x} = \Xi\Theta(X, U)$
 1: Construct a library of candidate functions $\Theta(X, U)$
 2: Solve the sparse regression problem $\dot{X} = \Theta(X, U)\Xi$ using a sparsity-promoting technique (e.g., LASSO)
 3: Identify the active terms and coefficients in $\Xi$ that form the model

---

2. Construct a library of nonlinear candidate functions $\Theta(X, U)$ of size $m \times q$, where $q$ is the number of candidate functions. These functions can be constant, polynomial, trigonometric, or more exotic functions of the $x$ and $u$.

3. Solve the sparse regression problem $\dot{X} = \Theta(X, U)\Xi$, where $\Xi$ is a matrix of coefficients of size $q \times n$, by minimizing the objective function

$$\min_{\Xi} \|\dot{X} - \Theta(X, U)\Xi\|_F^2 + \lambda\|\Xi\|_1, \tag{A1}$$

where $\|\cdot\|_F$ is the Frobenius norm, $\|\cdot\|_1$ is the $l_1$ norm, and $\lambda$ is a regularization parameter that controls the sparsity of $\Xi$.

4. Identify the sparse set of active terms in $\Theta(X, U)$ by selecting the rows of $\Xi$ that have nonzero entries. These terms form the governing equations of the dynamical system:

$$\dot{x} = f(x, u) = \sum_{i=1}^{q} \xi_i \theta_i(x, u), \tag{A2}$$

where $\xi_i$ is the i-th row of $\Xi$ and $\theta_i(x, u)$ is the i-th candidate function.

*Appendix A.2. Classical SINDY (MATLAB Code)*

**Listing A1.** MATLAB function for the classical SINDY.

```matlab
function Xi = sparsifyDynamics(Theta,dXdt,lambda,n)
% Copyright 2015, All Rights Reserved
% Code by Steven L. Brunton
% For Paper, "Discovering Governing Equations from Data:
%          Sparse Identification of Nonlinear Dynamical Systems"
% by S. L. Brunton, J. L. Proctor, and J. N. Kutz

% compute Sparse regression: sequential least squares
Xi = Theta\dXdt;  % initial guess: Least-squares

% lambda is our sparsification knob.
for k=1:10
    smallinds = (abs(Xi)<lambda);   % find small coefficients
    Xi(smallinds)=0;                % and threshold
    for ind = 1:n                   % n is state dimension
        biginds = ~smallinds(:,ind);
        % Regress dynamics onto remaining terms to find sparse Xi
        Xi(biginds,ind) = Theta(:,biginds)\dXdt(:,ind);
    end
end
```

*Appendix A.3. GenesisXi Algorithm*

The following is the description of Algorithm A2:

1. Given the vector $\Theta_L \in \mathbf{R}^q$, whose elements contain the number of the differential equation where the corresponding candidate function should be located, and $n$, which

is the dimension of the nonlinear system to be estimated, the matrix $\Xi_0 \in \mathbf{R}^{q \times n}$ is set to zero. For this case, $\Theta_L = [9\ 1\ 2\ 3\ 6\ 5\ 4\ 7\ 8\ 9\ 10\ 10\ 11\ 11\ 12\ 12]$.

2. The vector $\Theta_L$ is traversed from its first element to the last using the index $k$, simultaneously placing a "1" at the position $(k, \Theta_L(k))$ in the matrix $\Xi_0$.

---

**Algorithm A2** GenesisXi algorithm

---

**Require:** Data vector $\Theta_L$ and $n$
**Ensure:** Initial matrix $\Xi_0$
1: $\Xi_0 \leftarrow 0_{[q,n]}$
2: **for** $k = 1$ **to** $q$ **do**
3: $\quad \Xi_0(k, \Theta_L(k)) \leftarrow 1$
4: **end for**

---

*Appendix A.4. GenesisXi (Matlab Code)*

**Listing A2.** MATLAB function for the GenesisXi algorithm.

```matlab
function Xi_0 = GenesisXi(Theta_L,n)
% Construction of Xi_0

q=size(Theta_L,1); % Assuming that Theta_L is a column vector.
Xi_0=zeros(q,n);
for k=1:q
    Xi_0(k,Theta_l(k))=1;
end
```

*Appendix A.5. Modified SINDY (Matlab Code)*

**Listing A3.** MATLAB function with modifications to classical SINDY.

```matlab
function Xi = sparsifyDynamics_mod(Theta,dXdt,lambda,n,Xi0)
% Modified SINDY

Xi = Theta\dXdt;
Xi=Xi.*Xi0;   %Modification
for k=1:10
    smallinds = (abs(Xi)<lambda);
    Xi(smallinds)=0;
    for ind = 1:n
        biginds = ~smallinds(:,ind);
        Xi(biginds,ind) = Theta(:,biginds)\dXdt(:,ind);
    end
end
```

Roughly speaking, the modifications consist of removing the functions that wrongly appear in the differential equations after the first guess of the matrix $\Xi$ in line 4. To achieve this, the MATLAB operator ".∗" is applied to multiply each element of the matrix $\Xi$ obtained in line 4 by the corresponding element of the matrix $\Xi_0$. In other words, $\Xi_0$ is used as a template for the desired matrix $\Xi$. The remainder of the program follows exactly the structure of the classical SINDY function presented in Listing A2.

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
