# Peer review of "Identification of High-Order Nonlinear Coupled Systems Using a Data-Driven Approach"

_applsci, doi:10.3390/app14093864_

Round 1

Reviewer 1 Report

Comments and Suggestions for Authors

The paper describes a modification of a sindy algorithm to be applicable for dynamical systems. 

The introduction of the paper shows the motivation of the topic, the relevant literature and the novelty of the paper.

What was strange that the novelty description is organized into a different section, I think it not necessary, it would be better if this part can be the text of the original descriptions.

There are some minor misstakes corrected in the paper:

- the introduction text contains some inline equations where the role of each character should be explained

- the role of these characters explained later, this table maybe put before this text

- the explanation of the advantages/disadvantages can be detailed in the text and decrease the similiraties between the similar papaers.

- the conclusion can show something more about further development, applicability of the method

Reviewer 2 Report

Comments and Suggestions for Authors

The manuscript "Identification of high-order nonlinear coupled systems using a data-driven approach" introduces a method for identifying mathematical nonlinear systems, with a particular focus on high-order systems with coupled dynamics, as exemplified by a commercial quadcopter.

The author addresses a common misconception in the field and proposes a variant of the Sparse Identification of Nonlinear Dynamics (SINDY) approach to address it. However, the manuscript lacks specific details regarding the variant of the SINDY approach and how it differs from existing methods. The discussion section is also organized by topics and is quite concise. Additionally, more information regarding the experimental setup and validation of the proposed method would enhance the manuscript's credibility.

I recommend revisiting the existing literature and better defining the unsolved challenges of the area. From a methodological standpoint, I suggest including algorithmic details in the appendix rather than interspersing them throughout the main text. Furthermore, I believe that the scope of the project may not align perfectly with Applied Physics. It might be beneficial to submit the manuscript to a journal of the Computer Science area.

Comments on the Quality of English Language

The English of the whole manuscript should be revised

Reviewer 3 Report

Comments and Suggestions for Authors

This work introduces a modification to the classical SINDY algorithm, enabling the 376 identification of the mathematical model for high-order nonlinear systems with coupled 377 dynamics. The topic is interesting. Then, some key issues need the author's attention.

1.       The contribution should be further clarified from different aspects such as the novelty of the model under consideration, the significance of the proposed techniques and so forth.

2.       The author needs to further explain the innovation, including theoretical analysis and simulation analysis. The necessary remark needs to be increased.

3.       The authors are advised to polish the language carefully as there are some typos and grammatical errors throughout the paper.

4.       If possible, the necessary and detailed comparative analysis needs to be added.

Comments on the Quality of English Language

This work introduces a modification to the classical SINDY algorithm, enabling the 376 identification of the mathematical model for high-order nonlinear systems with coupled 377 dynamics. The topic is interesting. Then, some key issues need the author's attention.

1.       The contribution should be further clarified from different aspects such as the novelty of the model under consideration, the significance of the proposed techniques and so forth.

2.       The author needs to further explain the innovation, including theoretical analysis and simulation analysis. The necessary remark needs to be increased.

3.       The authors are advised to polish the language carefully as there are some typos and grammatical errors throughout the paper.

4.       If possible, the necessary and detailed comparative analysis needs to be added.

Round 2

Reviewer 2 Report

Comments and Suggestions for Authors

The authors significantly improved the manuscript by providing detailed clarification of the work's contribution in the introduction, expanding on the calculations performed, and notably, including an appendix with the code used for the simulations, allowing readers to replicate the simulations if desired. I believe the article is now at the desired level for Applied Sciences.

Comments on the Quality of English Language

No comments.